# Fully Automated DCNN-Based Thermal Images Annotation Using Neural Network Pretrained on RGB Data

**DOI:** 10.3390/s21041552

**Published:** 2021-02-23

**Authors:** Adam Ligocki, Ales Jelinek, Ludek Zalud, Esa Rahtu

**Affiliations:** 1Robotics and AI Research Group, Faculty of Electrical Engineering, Brno University of Technology, 61600 Brno, Czech Republic; zalud@feec.vutbr.cz; 2Cybernetics and Robotics Research Group, Central European Institute of Technology, Brno University of Technology, 61600 Brno, Czech Republic; ales.jelinek@vutbr.cz; 3Artificial Intelligence and Vision Research Group, Department of Computer Science, Tampere University, 33101 Tampere, Finland; esa.rahtu@tuni.fi

**Keywords:** deep convolutional neural networks, transfer learning, YOLO, RGB, IR, thermal, data annotation, object detector

## Abstract

One of the biggest challenges of training deep neural network is the need for massive data annotation. To train the neural network for object detection, millions of annotated training images are required. However, currently, there are no large-scale thermal image datasets that could be used to train the state of the art neural networks, while voluminous RGB image datasets are available. This paper presents a method that allows to create hundreds of thousands of annotated thermal images using the RGB pre-trained object detector. A dataset created in this way can be used to train object detectors with improved performance. The main gain of this work is the novel method for fully automatic thermal image labeling. The proposed system uses the RGB camera, thermal camera, 3D LiDAR, and the pre-trained neural network that detects objects in the RGB domain. Using this setup, it is possible to run the fully automated process that annotates the thermal images and creates the automatically annotated thermal training dataset. As the result, we created a dataset containing hundreds of thousands of annotated objects. This approach allows to train deep learning models with similar performance as the common human-annotation-based methods do. This paper also proposes several improvements to fine-tune the results with minimal human intervention. Finally, the evaluation of the proposed solution shows that the method gives significantly better results than training the neural network with standard small-scale hand-annotated thermal image datasets.

## 1. Introduction

### 1.1. Our Research

These days, the Advanced driver-assistance systems (ADAS) systems and autonomous driving are the edge breaking segments of industry. Car producers invest extreme resources into the systems that would improve security or add more comfort to the driving experience. As the developers equip vehicles with better means to sense and understand their surroundings, neural networks are more often used. This paper focuses on training neural networks to detect objects in thermal images, respectively, creating a thermal image dataset used for a neural network in a cheaper and more automated way than the common human-based annotation approach. The thermal data are very important for modern ADAS systems as the common RGB sensors cannot detect objects in bad weather or light conditions [1]. In this case, thermal imaging gives a handy tool to detect many types of objects, especially those with a higher temperature above the background, like all living creatures of running vehicles.

Today, the state-of-the-art-data processing techniques are the deep convolutional neural networks [2,3], which completely changed how we think about the data processing, especially for the computer vision domain.

Ten years ago, the best results had knowledge-driven approaches, where scientists or developers had to define the data processing pipeline rules. Today, the best scores in the publicly available data processing challenges are given by the data-driven deep convolutional neural network models [4,5,6].

However, one of the huge problems of those data-driven approaches is the need for an enormous amount of data that is necessary to make it possible to understand and to generalize the problem by a neural network. We speak about hundreds of thousands, or millions, of annotated image-label pairs that today’s state-of-the-art architectures need to generalize a problem.

To our best knowledge, there is no publicly available dataset large-enough to reliably train a detector for bad weather or light conditions. This was our primary motivation for developing an automated technique for creation of those datasets since traditional approach is extremely time-demanding and consumes a large amount of human effort and money. The RGB object detection datasets are quite well-covered in science, and some projects provide even millions of RGB annotated images [7]. In this paper, we present a technique to leverage the usability of these valuable resources and harness them in IR image processing, as well. Figure 1 shows the overall idea of transferring detection from RGB to IR image.

### 1.2. Related Work

Our research that preceded this paper is covered in the previously presented Brno Urban Dataset (BUD) [8], that covers the original multiple-sensor dataset recording and the sensory framework setup, and the Atlas Fusion whitepaper [9], which describes the software used for automatical data annotation.

There are only a few thermal image datasets available. Their key properties are as follows:

The one that best suits our requirements is the FLIR dataset, which is currently available in three versions. The fully free FLIR Starter Dataset [10], and two extensions, the Enhanced San Francisco Dataset [11], and the Enhanced EU Dataset [12], that could be accessed on request. The basic version contains 14,000 pictures, and, counting all the extensions, the FLIR provides a set of 38,000 images. However, the annotated classes are not consistent in all of the datasets, and the extensions do not add any new annotations of the most problematic and low-represented class of animals.

The FLIR dataset is a handy tool for evaluating performance or it could be used for transfer learning [13] because it is very precisely annotated. But this dataset does not fulfill the requirement of the hundreds of thousands of images, and it puts high requirements on human labor to extend the dataset in the future.

The second suitable example is the KAIST dataset [14]. The authors created a device that allowed them to put the optical centres of both an RGB and a thermal camera into the same place, making it easy to later scale automatic annotation by the YOLO neural network, which is similar to our approach. It provides close to one hundred thousand images, but there are annotated only with pedestrians. A similar example could be Reference [15], that also annotates only pedestrians, and provides only about 4000 images.

From the bigger datasets, we can mention the LSI dataset [16] (15,000 images), ASL-TID [17] (6000 images), and the OSU-CT dataset [18] (17,000 images). The TIV benchmark dataset [19] (60,000 images) also contains the annotations of cars, bikes and motorbikes, but only some part of data are the outdoor traffic data. There are also indoor scenes. The problem of all these works is that they mainly focus on humans and that there is a small representation of other real-traffic classes. We could merge all those datasets and unify their annotation style, but we would reach only slightly above one hundred thousand images in the best case, with very few animals, vehicles, and bikes and other real-traffic annotations.

On the other hand, the well known large scale datasets, like KITTI [20], the Berkeley DeepDrive dataset [21], Apollo Space dataset [22], or the Oxford RobotCar dataset [23] do not contain any thermal image data.

By using the method mentioned before in the KAIST dataset paper [14], authors can rectify images and directly map detection from the RGB to the thermal image. A similar technique is used by [24]. This is a very clever and straightforward method that can be easily extended with new annotated classes and more images in the future. However, the method requires specialized hardware and a non-trivial setup, which is not widely available.

Another category are the synthetic thermal images generated by the Generative Adversarial Networks (GANs) [25]. Methods, like References [26,27], use the pre-trained deep convolutional networks to convert RGB images into the pseudo-thermal-like images, so the generated pseudo-thermal image covers the same scene as the RGB does. Later, these methods perform the object detection process on the original RGB images using the YOLO network and annotations are then finally be used with the GAN-generated pseudo thermal image as a ground truth. But these methods introduce vast bias to the created dataset as the generated pixel intensities of the objects do not correspond with real-world data.

There are also propositions to use the networks pre-trained on RGB images to directly perform object detection on the thermal images [28], but this approach does not provide any benefit over the knowledge comprised in the neural network’s weights.

All other methods that we found in the available literature are dealing with single-class (mostly pedestrians) detections [29,30,31,32] and do not scale the training dataset above the limit of several hundreds or a few thousand images.

In this paper, we present a method that uses the RGB camera, thermal camera, and 3D LiDAR sensor to automatically generate the annotated thermal image dataset that counts hundreds of thousands annotated images. A fully automatically created dataset gives comparable results in the neural network training process as the human-annotated datasets do, which saves a large amount of human labor and is easily scalable if new classes or training data are going to be added in the future. By adding small human intervention in dataset creation, we improved the training process, so it outperforms training neural networks on these days’ largest human-annotated datasets.

## 2. Datasets

In this paper, we use two different datasets to validate results of our work. The first one, the FLIR ADAS dataset (FLIR), is a human-annotated thermal image dataset used as a reference one. The second one is the Brno Urban Dataset (BUD), which we used to create the automatically annotated thermal image dataset. Later, we trained neural networks on both datasets, the hand-annotated and the automatically annotated, and compared their performance on the test set that was common for all models.

### 2.1. FLIR ADAS Dataset

The FLIR Thermal dataset [10] is fully free and documented set of about fourteen thousand of human-annotated images, captured by the FLIR Tau 2 infra-red camera. The free version of the dataset contains the bounding box annotations of approximately forty thousand people, fifty thousand vehicles, over five thousand bikes and motorbikes, two hundred and forty dogs, and nearly three thousand other vehicles, in daylight and night time conditions.

Every single thermal camera frame is stored in the dataset as a 14-bit 512 × 640 px image in tiff format and a common grayscale 8-bit format with the same resolution. In addition, at the same time, each IR frame is accompanied by a time-synchronized RGB image, taken by a nearby RGB camera. However, we are not using these RGB images in this work.

The dataset is divided into three logical parts (for details see Table 1), the training one (eight thousand thermal images), the validation one with approximately two thousand IR frames, and the video section containing nearly four thousand images in the video sequences recorded at 30 frames per second.

The non-free version of the dataset extends the number of images and the linked annotations approximately three times. The extended Enhanced San Francisco dataset [11] and the FLIR European Regional Thermal Dataset [11] provide images captured in foggy and rainy conditions.

### 2.2. Brno Urban Dataset

The Brno Urban Dataset [8] is a result of work in our research group. It is the basis for the method that we present in this paper. The dataset contains over 350 km, ten hours of everyday real-life traffic situations recorded by a car in the middle-size city of Brno. Data samples are shown in Figure 2. The dataset provides the raw data from four RGB cameras with the 1920 × 1200 px resolution. Two frontal and two lateral sensors cover approximately 220 degrees of view around the car. The single FLIR Tau 2 thermal camera with 512 × 640 pixel resolution faces the same direction as the two frontal RGB cameras. The RGB cameras are recording at a rate of 10 frames per second (fps), and the thermal camera records at 30 fps. Further follows two 3D LiDARs with 32 lasers each, which provide about 1.2 million points per second in 10 scans per second. The dataset also provides the 3D linear acceleration data, angular velocity, and the magnetic field intensity in all the axes measured by the Xsens inertial measurement unit and the RTK GNSS receiver with differential antennas and heading vector estimation. Sensor overview is in Table 2, and the layout of the frame is in Figure 3.

For every sensor data stream in the dataset, a dedicated text file keeps timestamps for all data packets with millisecond precision. The precise data timing allows us to create reliable inter-sensory fusion.

The Brno Urban Dataset is fully open-source, and we encourage everyone to visit the project’s GitHub web page at Reference [33].

## 3. Software Backend

### 3.1. YOLOv5

As mentioned before, we utilize a state-of-the-art neural network that detects objects in the RGB images. We used an existing open-source project published by Ultralytics, available in Reference [34].

This framework provides the API to run the detection in the existing RGB images. It gives a straightforward and lightweight tool to train neural networks on our data and apply the transfer learning techniques.

In addition, the framework provides four architecture configurations of the neural network, ranging from the straightforward and lightweight one (version S), where the training on our dataset took us only several hours, up to the complex architecture (version X), which took a few weeks to train and test our method’s results.

### 3.2. YOLO Architecture Principles

The YOLO (You Only Look Once) neural network architecture was initially published in [35]. Compared, for example, to the R-CNN [36] algorithm, which is based on region proposing, the YOLO architecture provides detection during a single image pass through the network. This feature allows authors to create an end-to-end learning pipeline and very fast inference.

Originally, YOLOv1 was splitting the image into an SxS cell-grid where, for every cell, two bounding boxes are predicted (see Figure 4). Each bounding box was defined by a X and Y position, width and height, and object confidence. Above that, for all bounding boxes in every cell, there was a class confidence vector.

Summing it up, the YOLOv1 has a classic convolutional neural network backend that extracts feature maps from the input image. The fully connected layers later use these feature maps to generate the output tensor. At the output of neural network, there is a S∗S∗(noOfBoundingBoxex∗5+noOfDetectedClasses) tensor, that represents output detections. Originally, for YOLOv1, output tensor dimensions were 7∗7∗(2∗5+20).

YOLOv2 [37] introduced several improvements like batch normalization layers, multi-scale training, the bounding boxes with dimension priors, or the new neural network backend called Darknet-19. Those partial improvements moved the original performance of YOLOv1 from 63.4 mean average precision (mAP) up to 78.6 mAP on the second version on the VOC2007 dataset.

The last version is YOLOv3 where the authors introduce several other improvements. The most significant one is, once again, the new backend convolutional neural network called Darknet-53. The output classifier does not contain softmax on the output, so the network can classify the object into multiple classes (i.e., Woman and Person). Another improvement is a pyramid feature network, which means the network does not predict bounding boxes for a single S*S cell-grid. Instead, in case of YOLOv3, the network is using three different cell-grids. That allows the network to detect tiny objects that cover a small area of the image, as well as the objects that take the entire image space. In addition, the YOLOv3 estimates the class probability for every bounding box separately.

In our case, the YOLOv5 neural network gives three different output tensors. Each tensor is specialized in detecting objects of different sizes. For the 640 × 640 × 3 input image, our network provides tensors of a 3 × 20 × 20 × (N + 5), 3 × 40 × 40 × (N + 5) and 3 × 80 × 80 × (N + 5) dimensions on the output. The 3 in the expressions represents the number of anchor boxes predicted per cell, the 20 × 20 (40 × 40 and 80 × 80) gives the dimensions of the cell-grids and N is the number of predicted classes (4 in our case). The 5 represents the bounding box X and Y position, width, height, and confidence of the detection. The visualization of the example can be seen in Figure 5.

The YOLOv5 is not a new and improved version of the YOLO architecture, but rather it is a newer implementation of YOLOv3 in the PyTorch framework [38].

### 3.3. Atlas Fusion

Atlas Fusion is a software developed by our research team. This project implements all the necessary algorithms used to map RGB detections into the thermal images.

First of all, the software combines IMU and RTK GNSS measurements into position and motion estimation. It combines data from multiple LiDARs captured in time into a single dense point cloud model. It also implements point cloud projection into a camera frame to create a depth map of the scene that the camera captures. Finally, it implements reprojection of pre-calculated object detections on RGB images into the thermal images. All the details of how the Atlas software is used can be found in the following section.

For a better overview, please see the whitepaper [9] that describes all the components of this software. This software is also open-source and fully available in Reference [39].

## 4. Generating Synthetic Dataset

We used two training datasets to compare the results of a neural network trained by a human-annotated FLIR dataset and our automatically annotated thermal image dataset.

The automatically annotated dataset is called BUDIR (Brno Urban Dataset—IR). To create it we applied the following three steps. The first one is aggregation of multiple LiDAR sensor data into a single dense point cloud model of the surrounding environment. The second step is object detection at the RGB images and estimation of their distance using the dense point cloud model. Finally, the objects detected in 3D are reprojected into the corresponding thermal camera images.

### 4.1. LiDAR Data Aggregation

We use the rotating 3D LiDAR scanners with 32 lasers with approximately 2000 points scanned by the sensor per rotation per laser. The sensor takes a single scan every 100 ms, which seems to be a lot of data. However, if we want to detect a 2-m wide car at a distance of 30 m, the scanner provides us with approximately ten measured points per laser on this object. Because of this, we need to aggregate measurements from multiple sensors for an extended period of time (more than 1 s, which corresponds to ten or more scans per LiDAR). This multiplies the number of measurements of a single object at least by one numeric order. See Figure 6.

Another problem is removal of the inevitable distortion of the LiDAR data. As the sensor moves during the scanning, the relative position to a scanned static object is changing.

Let us imagine a situation where the laser measures the distance of the object at the very front of the sensor. The first measurement says the object is at a distance of 10 m. Then, the scanner measures the entire surrounding, and, after 100 ms, it returns to the original orientation. If we assume that the car on which the sensor is mounted is moving 10 m/s forward, the final measurement differs by 1 m compared to the first one. The situation gets even worse if the car is not only performing the translation movement but rotates, as well [40,41].

We deal with both of these problems by using a single method. Thanks to the precise GNSS receiver and the IMU sensor, we have accurate information about the car’s position and rotation with an update frequency of 400 Hz. We can undistort the newly measured 3D point cloud by splitting it into N (100 in our case) point-batches. Every point batch contains the measurement from a strictly defined measurement period. As we know the time when the points were scanned, we can backtrace the car’s position at this time, and transform the points (see Equation (Equation 1)) using the homogeneous transformation from sensor’s coordination frame into the global frame in which the entire 3D model of the surrounding is constructed. This way, every sensor’s scan is undistorted and transformed into the unified coordinate system. All the undistorted data is aggregated into a single point cloud model. The point batches older than the required history are removed from the memory.
(1)$$Pworld=TF_{car2world}*TF_{imu2car}*TF_{sen2imu}*Psen,$$
where P represents the point batches, and TFs represent the homogeneous transformations between the coordinate frames.

### 4.2. 3D Detection in the RGB Image

In the second step, we use the pre-trained neural network to process all of the approximately 300,000 collected RGB images and perform object detection on them. This way, the neural network annotates all the RGB images. This approach is not entirely reliable and makes more mistakes during the annotation process than a human annotator would make, although most annotated objects are annotated correctly. For now, we expect that the student neural network (network, that will be trained on automatically annotated thermal images) will be able to generalize the problem and will not learn the mistakes that the teacher network (the network that annotates RGB images) made.

Further, for every detected object, we reproject the environment’s point cloud model into the RGB image plane, which creates a sparse depth map for given image (see Figure 7).

We know the 3D point cloud model of the vehicle’s surrounding in world coordinates (Equation (Equation 1)) and also the position and orientation of the car and all its sensors. This gives us enough information to reproject the points from the 3D dense point cloud model of the surroundings onto the camera frame using Equation (Equation 2). The equation does not involve lens distortion correction. After that, we extract the points inside the detected object’s bounding boxes, and by taking the median of all the extracted point distances, we estimate the distance of the object from the RGB camera frame (see Figure 8).
(2)$$\left[\begin{array}{c} u\\ v\\ 1 \end{array}\right]=\left[\begin{array}{ccc} f_{x} & 0 & c_{x}\\ 0 & f_{y} & c_{y}\\ 0 & 0 & 1 \end{array}\right]*\left[\begin{array}{cccc} r_{11} & r_{12} & r_{13} & t_{1}\\ r_{21} & r_{22} & r_{23} & t_{2}\\ r_{31} & r_{32} & r_{33} & t_{3} \end{array}\right]*\left[\begin{array}{c} x\\ y\\ z\\ 1 \end{array}\right],$$
where (Equation 2), *x*, *y*, and *z* are the world coordinates, $r_{ij}$ are the coefficients of the rotation matrix, $t_{i}$ are the translation coefficients, $f_{x}$ and $f_{y}$ are the focal length of camera’s lens, $c_{x}$ and $c_{y}$ are the camera principal point coordinates, and the *u* and *v* are the pixel coordinates on the image.

Now, we know the object’s 2D projection and direction in which the object is located with respect to the camera. We also know the distance of the detected object from the camera’s frame. The detection can be determined as a frustum in the 3D space.

### 4.3. Thermal Images Annotation

The last step is to take the corresponding thermal image to the RGB image that we processed and project the corners of the frustum’s front face onto the thermal image. This way, the final annotation of the captured thermal image is acquired that can later be used for training purposes.

As we mentioned before, not all automatically created annotations are entirely correct. However, the number of miss-detections or other annotation failures is a negligible minority in the entire set of annotated objects. We expect that the student neural network will be able to overcome the miss-annotation and learn only the valid detections. The same problem can be seen even in the case of human learning. Not everything that the teachers taught us is one hundred percent correct information, and, in the amount of received data, we can extract and learn the type of information that we see as the most reliable.

The overall process visualization is shown in Figure 1, Figure 9 and Figure 10.

### 4.4. Reducing No. of Classes

This paper deals with two different datasets and compares the results that we reached on each of them. To make a fair comparison, we have to ensure the same conditions for both of them. The FLIR dataset in the free version annotates pedestrians, vehicles, bikes (and motorbikes as one), and animals. The number of annotations for every class is shown in Table 1.

The teacher YOLOv5 neural network that was pre-trained on the COCO dataset [42] covers eighty different classes. This fact forced us to reduce the number of classes used during the learning process. From the original COCO class list, we used classes no. 1 (person) as a ’pedestrian,’ classes no. 2 (bicycle), and 4 (motorcycle) as a ’bike’, classes no. 3 (car), 6 (bus), 7 (train) and 8 (truck) as a ’vehicle’ and classes 15 (bird), 16 (car), 17 (dog), 18 (horse), 19 (sheep), 20 (cow), 21 (elephant), 22 (bear), 23 (zebra) and 14 (giraffe) as an ‘animal’, although it is quite rare to see elephants, zebras, or giraffes here in central Europe. Other types of detected objects were not used.

Table 3 shows the number of annotated objects in our automatically created dataset. Compared with numbers in Table 1, we created a nearly 15 times bigger dataset than the most commonly used free to download alternative. Several annotated images from both datasets are shown in Figure 11.

From Table 3, we can see that the generated dataset is imbalanced by the number of annotations of different classes. The vehicles to human ratio is approximately 4.7:1, vehicles to bikes ratio is 73.2:1, and for vehicles to animals ratio is an even higher 3352:1. If we dealt with an image classification task, it would be impossible to train the model in any reasonable way. As for an object detection task, the situation is better. The object detectors are not as sensitive to the dataset imbalance as classifiers are. One way is to use the augmentation techniques [43,44] and increase the number of training samples that contains low-represented classes in the training set. Another option is to modify the training cost function [45] and put more weight on these classes represented by a small number of samples.

In this work, we considered this problem, and we decided not to modify the original form of the BUDIR dataset, and, during the training process, we realized that it was not a significant problem for ’pedestrian’ and ’bike’ classes. The ’animal’ class was the only one that was problematic to train. This problem was common for both the BUDIR and FLIR dataset. We decided not to change the conditions and left both datasets unchanged.

## 5. Training Neural Networks on the New Dataset

This section describes the experiment of training neural networks on both the FLIR and BUDIR dataset and comparison of models performance.

### 5.1. Metrics

When we train artificial intelligence models, we need to measure their performance. In the field of object detection in images, the most common metric is the mean Average Precision (mAP), first introduced by PASCAL VOC [46] as an mAP (0.5). The 0.5 value means that the true positive detections are those for which the Intersection over Union (IoU) value between the detection and the ground truth bounding boxes is at least 0.5.

Another variant used in this paper is the mAP (0.5:0.95) (COCO’s [42] standard metric). It means we estimate the mAP for IoU levels from 0.5 up to 0.95 with a 0.05 step and calculate the average of all the partial mAPs.

Moreover, we measured also the precision, recall, and the derived metrics from both-the F1 score.

### 5.2. Methodology

The entire process of neural networks training and testing their performance is as follows. For both datasets, FLIR and the BUDIR, we created the training and validation subsets using an 85:15 ratio. The images were split into two subsets by 100-image chunks. We used splitting by chunks to avoid where there would be images captured with a small-time difference (the training and validation subsets would contain images with the nearly the same scene and the same objects).

The correct way to create the testing dataset is to pick up a set of annotated samples from the set of all data and exclude those test data from the training process. It guarantees that the final model performance measurement is statistically independent from the training process. This approach is the best solution if we have a large number of annotated data. In our case, there are two difficulties, the relatively small number of images in the FLIR dataset, only 14,000, and the tiny number of animals in both the BUDIR and the FLIR datasets. If we would split the data in the common ratios 70:15:15, training:validation:test subsets, we would significantly lower the number of annotations in all subsets. The problem would be even worse for animals as there are only several hundreds of objects of this class in each dataset. If we split animal annotations into three subsets, the number of representants in each subset would be deficient and untrustworthy. Thus, we decided not to split datasets into three subsets but rather to create the test set as a combination of validation sets from both the BUDIR and the FLIR datasets. Hence, the numbers of annotated objects in training and validation sets are as high as possible.

All the FLIR dataset images are adjusted to the constant dynamic range that provides a good level of contrast for all annotated objects. On the other hand, the BUDIR images have an adjusted dynamic range concerning the temperature extremes in the scene. We can see the effect in Figure 11, where the BUDIR images are much more diverse in contrast level. If we combine both datasets into a single test one, we can eliminate the score overfitting on a specific dynamic range of the images. If we combine both validation sets into a single test set, we can measure the performance for models trained on the different datasets in the same unified conditions for both cases.

### 5.3. Training Details

We trained all networks on the Nvidia GTX 1080 GPU card. The minimalistic variant was the YOLOv5 S model (7.3 mil training parameters) trained on the FLIR dataset, which took around 8 hours to finish (100 epochs). On the other hand, we spent five weeks training the YOLOv5 X (87.7 mil training parameters) model on the 300,000 images of the BUDIR dataset for the same number of epochs.

We trained neural networks on the images with 640 × 640 px resolution. The batch size was four images per batch. ADAM optimizer was used, and the training lasted for 100 epochs.

Additionally, we used the YOLOv5 feature to augment training images by changing scale, dynamic range, and cropping images during the training process.

### 5.4. Training from Scratch

Initially, we trained the “S” and “X” versions of the YOLOv5 neural network on both FLIR and BUDIR datasets. As the best representative of the entire training procedure, we always took the weights with the best results on the validation set.

On both BUDIR trained neural networks, approximately in 23rd and 53rd epochs (see Figure 12), the score of BUDIR-trained neural networks is dramatically growing. These are the moments when the neural networks started to recognize animals. However, when we tried to extend both FLIR training processes up to 300 epochs, the validation scores started to fall slightly after the 100th epoch, and the FLIR-trained neural networks were never able to recognize animals on a significant level. It rather overfits.

The performance results on the combined test dataset are shown in Table 4, and the performance of the trained neural networks during the training process is shown in Figure 13. As we can see, the FLIR-trained models are better at recognizing pedestrians and bikes. On the other hand, BUDIR-trained neural networks outperform FLIR-trained in vehicle and animal detection. Summing it up, the BUDIR-trained models have slightly better average mAP (0.5), mAP (0.5:0.95), F1, and precision scores. The FLIR-trained models are doing better from the recall point of view.

The visualization of several images with object detection by all four trained models can be found in Figure 14. Here, it is quite clear that the FLIR-trained neural networks perform better in detecting the pedestrians and the BUDIR-trained neural networks have notable distortion of the bounding boxes that cover detected objects. The non-perfect bounding boxes fitting as the training data contains this type of distortion by the RGB to IR projection. On the other hand, BUDIR-trained neural networks have better results when detecting smaller objects, animals, and objects in lower contrast images.

### 5.5. Transfer Learning

In the second step, we tried to use the pre-trained weights from the original RGB oriented model. As the neural network performs detection over the image, the inner convolutional layers work as simple geometrical feature detectors. The deeper we go through the network, the more complex geometrical shapes are detected by the convolutional kernels. If we think about the RGB and thermal image object detections, the model does the very same work in both cases. From this point of view, there is no need to train the network from scratch, but we can use the existing detectors and tune them up for the thermal images. This technique is called transfer learning [13].

We tried to train each combination of the “S” and “X” YOLOv5 neural network on the FLIR and BUDIR datasets, but this time we used the pre-trained weights to verify the model’s object detection performance with the previous training-from-scratch approach. In the end, we once again tested all the trained neural networks on the validation dataset, which contains randomly selected one thousand images from the FLIR dataset validation set and one thousand images from the BUDIR validation set. The results are shown in Table 5. Moreover, we added a comparison with the original RGB-trained neural network.

From the results, it is clear that transfer learning gives better results than the training-from-scratch. In addition, for FLIR-trained neural networks, the improvement is more significant than for BUDIR-trained neural networks. We explain it as proof that, during the training, the neural networks better generalized the problem when training on a much larger BUDIR dataset and could not generalize that well using the smaller but better annotated FLIR dataset.

## 6. Fine-Tuning the Results

As was already mentioned, the automatically generated dataset is much bigger, and the model is able to learn longer without over-fitting, which makes it more robust. On the other hand, the annotations are sometimes misplaced, or the dimensions of the annotations are not precisely aligned with the objects. The effect of this is that the output detections from the neural network trained on the BUDIR dataset are slightly underfitting the real dimension of the detected object. It was the main reason causing worse results than with the FLIR-trained neural networks in the detection of pedestrians and bikes. Objects of these two classes are relatively small compared to vehicles, and even a small misplacement has much bigger effect on the model’s final performance.

We tried to solve this issue by fine-tuning the neural network trained on the BUDIR dataset by letting it learn for a few more epochs (ten epochs) on a small amount of data for which we manually fixed the localization of the annotation bounding boxes. Combination of the training on a large scale BUDIR dataset and later fine-tuning results on a small amount of human-annotated images, we expect better and more robust results.

To prepare the fine-tuning training dataset, we merged both the FLIR and the BUDIR training dataset and selected 75 images from the training dataset that contained at least one animal or one bike. Another 75 images were selected randomly from the same training set. For each of those 150 images, we fixed the existing data annotation bugs. We then once again tried to train the transfer-learned YOLOv5x that was previously trained on both the BUDIR and FLIR datasets. The final results are listed in Table 6, and the visualizations are shown in Figure 14.

Table 6 shows that the neural networks trained on the BUDIR from scratch or by transfer learning using the weights pre-trained on the RGB data give similar performance results compared to the models trained on the FLIR dataset. After applying the finetuning, the performance of FLIR-trained models changed only negligibly. On the other hand, the BUDIR-trained models’ performance has grown significantly, and the finetuned BUDIR-trained YOLOv5x outperforms all others.

## 7. Experiment Reproducibility

This chapter briefly recapitulates all the steps that are needed to reproduce our experiment.

First it is necessary to download our Brno Urban Dataset (BUD) [8] from Reference [33] and the Atlas Fusion software [9] from Reference [39]. The Atlas Fusion uses the BUD to generate an annotated thermal image dataset in the format compatible with the later used neural network’s framework.

The quality of generated dataset (BUDIR) could be validated by the neural network training on this data. In our work, we used Ultralitycs YOLOv5 framework available from [34]. We split dataset into two subsets, the training and the validation one, by a ratio of 85:15.

## 8. Discussion

This paper proposed and tested a new method for creation of large scale annotated thermal image datasets using a pre-trained deep convolutional neural network. We employed publically available datasets with a large amount of recorded RGB and thermal images and LiDAR data. These were used in the process of automatic annotation of RGB data and reprojection of the detections into the thermal images via the depth map of the scene that we created using the LiDAR data. In this way, we created a dataset of 300,000 annotated images for the thermal image object detection purposes.

While methods in other similar works, such as Reference [14] or Reference [24], allow annotation of thermal images in much easier way and without any parallax distortion, they require specialized hardware. In comparison, our method is universal and could be applied to any calibrated hardware layout without special mechanical constructions. Additionally, our method also generates the depth scene data as an additional benefit.

Another example of a convolutional neural network-based thermal image annotation is Reference [28], but the authors focus only on a single class and do not provide any number-based evaluation of their approach.

To test the quality of our dataset, we compared the results of training several neural network models on both our and the FLIR dataset.

In this phase, we found that neural networks trained on the FLIR dataset (on a smaller, but more precisely human-annotated data) were better in detecting humans and bikes. On the other hand, the models trained on the large scale BUDIR dataset were better in detecting vehicles and animals. The overall average score for all classes was similar for both the BUDIR and FLIR trained models. It shows that our method proves itself, and it is a relevant method that can replace the costly human-annotation process and provide very similar results.

In the second phase, we tried to improve the process by adding a small amount of human work to fine-tune the trained neural networks and remove distortion problems that neural networks trained from the automatically annotated data.

We hand-annotated 150 selected images that contained all detected classes and let the models fine-tune their bounding box dimensions estimation. In this way, we significantly improved the BUDIR-trained models’ score, which was affected by the distortion problem, while the fine-tuning has only a negligible effect on the FLIR-trained models.

The second phase proved that using an object detector trained in a different part of the light spectrum could very effectively replace most of the costly process of annotating thermal images for object detection training in the thermal domain. The most important results are summarized in Figure 14 and Table 6. By applying the finetuning method, the BUDIR-trained models outperform the FLIR-trained models in all detected classes, as well as in the overall score.

By developing this method, we created a dataset larger than all other publicly available datasets combined. Moreover, we annotated four real-traffic classes in a unified style, when most of the available sets annotate only pedestrians.

To compare the time consumed by the annotation processes, a typical desktop computer processed 10 h of recorded data and annotated 300,000 thermal images in 24 h. The hand annotation of 150 images took us about one hour of work. Working with the same speed, annotation of 300,000 mages would take us about 2000 h.

In the future, our method could be extended with larger number of annotated classes. We are also interested in studying if the method could be used to create a similar dataset to train object detection in the LiDAR data, respectively, in the depth maps.

## 9. Conclusions

This paper proposes and validates automatic annotation process for thermal images by using the RGB and thermal cameras combined with a LiDAR sensor. We created the dataset of 300,000 annotated images and tested the dataset quality in the neural network training process. The results show that the models trained on a large scale dataset created by our method give better thermal image object detection results (0.661 mAP (0.5)) compared to the models trained on significantly smaller and publicly available hand-annotated data (0.555 mAP (0.5)).

## Figures and Tables

**Figure 1 sensors-21-01552-f001:**
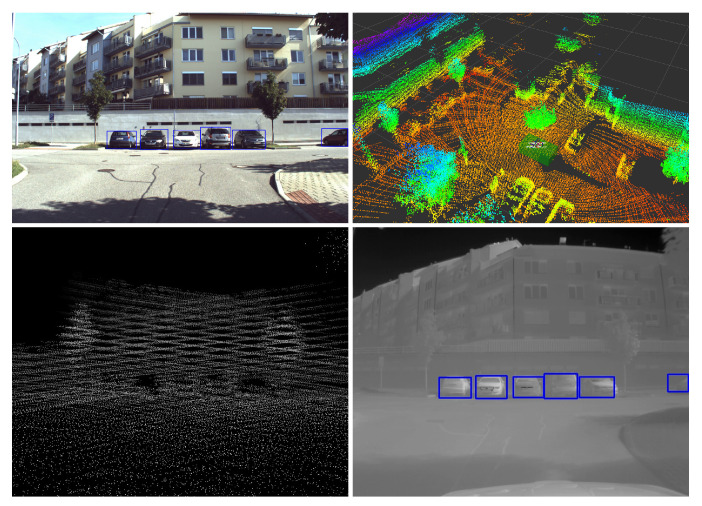
The image shows the basic idea of projecting detections from the RGB image via the depth map into the thermal image. **Top left**—RGB image with detected objects; **top right**—aggregated 3D dense point cloud model of the surrounding environment of the car; **bottom left**—point cloud projected into the image plane; **bottom right**—thermal image with detections transferred from RGB one.

**Figure 2 sensors-21-01552-f002:**
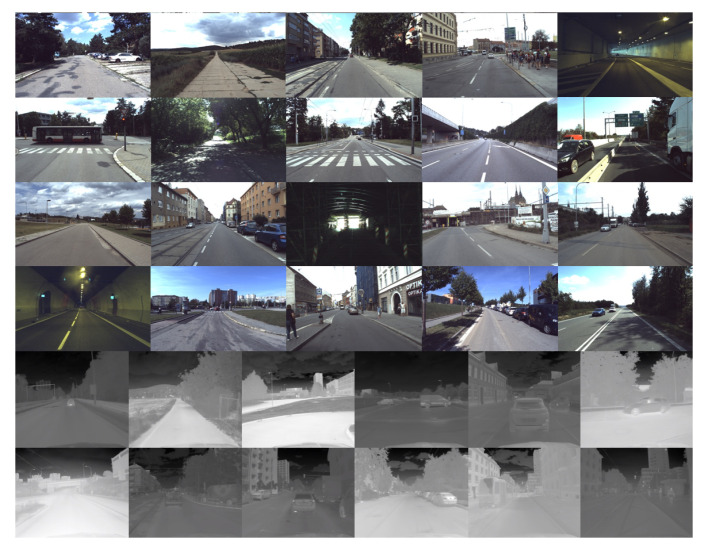
The image shows a brief overview of data captured by the left frontal RGB camera and the thermal camera in the Brno Urban Dataset (BUD). These two sensors are the most important in this paper. We are using the RGB camera images as a source for the RGB neural network that generates RGB object detections. These detections are later mapped into the thermal images. In this way, we create the IR training dataset.

**Figure 3 sensors-21-01552-f003:**
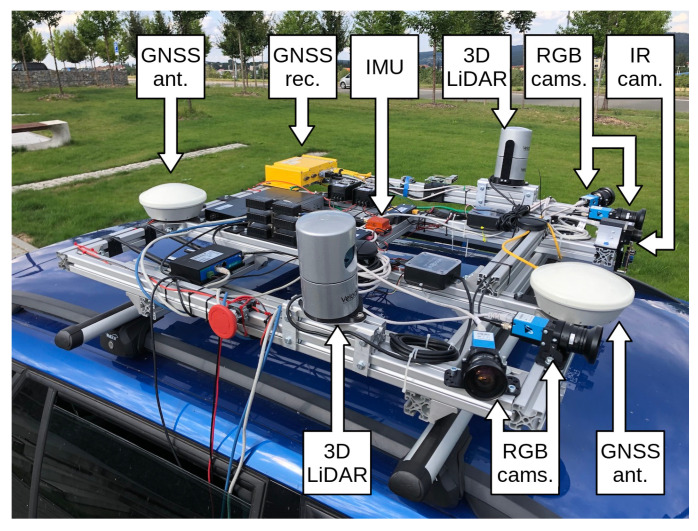
Sensor setup overview. The left frontal RGB camera is the closest one to the thermal camera sensor. Data from these two cameras are the main data sources used in work described by this paper. First, 3D LiDAR takes place several dozens of centimeters behind the cameras. The second sensor is placed on the opposite side of the frame.

**Figure 4 sensors-21-01552-f004:**
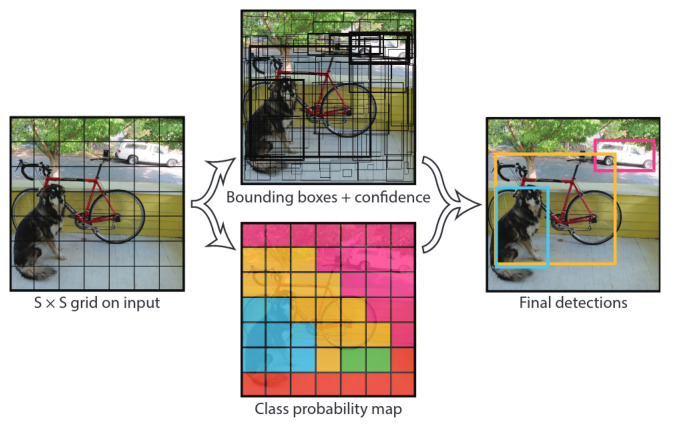
Ref. [35]: Original YOLOv1 architecture predicts class for every cell in the 7∗7 grid. In addition, every cell predicts two possible bounding boxes and their confidence. Merging these two pieces of information creates the output object detection.

**Figure 5 sensors-21-01552-f005:**
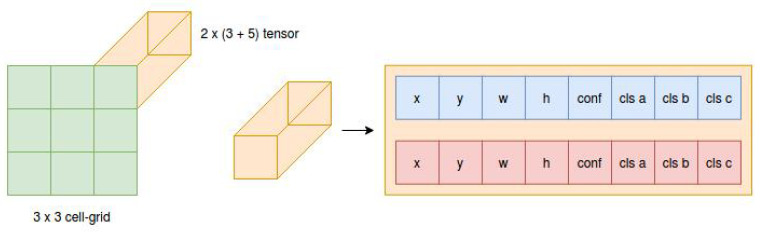
A simple example of the single output tensor that represents the 3 × 3 cell-grid where each cell predicts two bounding boxes of given X and Y position, width W and height H, all relative to the cell area and position, the bounding box detection confidence, and three probabilities of object affiliation to the given classes.

**Figure 6 sensors-21-01552-f006:**
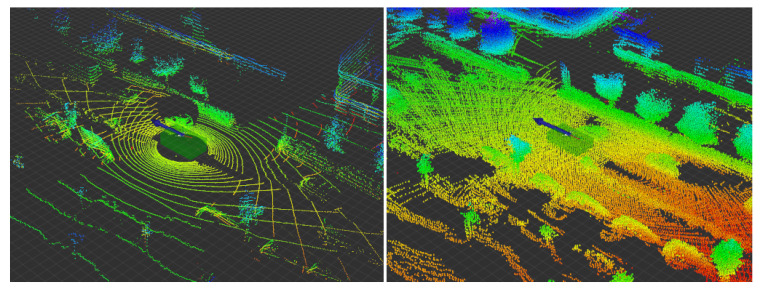
Visualization of the non-aggregated point cloud from two Velodyne HDL-32e scanners (**left**) and the aggregated point cloud over the 1.5-s period (**right**).

**Figure 7 sensors-21-01552-f007:**
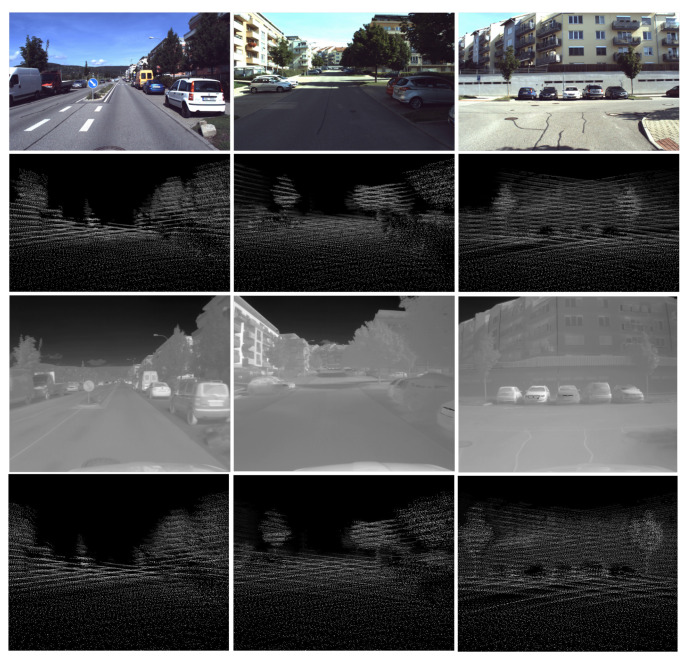
Example of a depth map created by reprojection of point cloud data into the RGB and thermal camera frames. First row—RGB image; second row—corresponding depth map; third row-thermal image of the same scene; forth row—depth map projected into the thermal camera frame.

**Figure 8 sensors-21-01552-f008:**
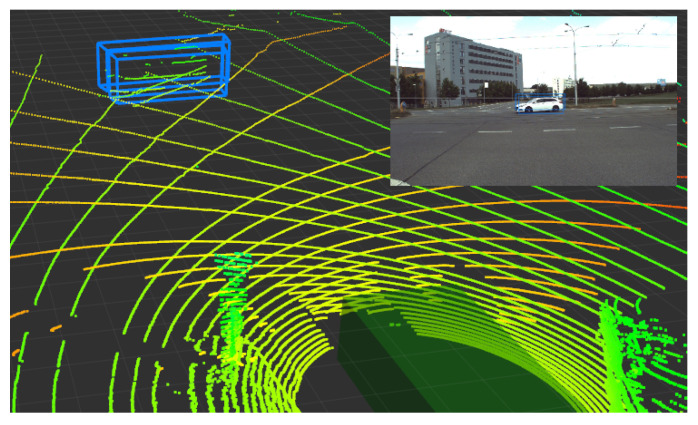
Example of a car object detected in the image. By projecting LiDAR measurements, we estimate the distance of the object in the 3D space with respect to the sensors. The 3D detection is then represented as a truncated frustum.

**Figure 9 sensors-21-01552-f009:**
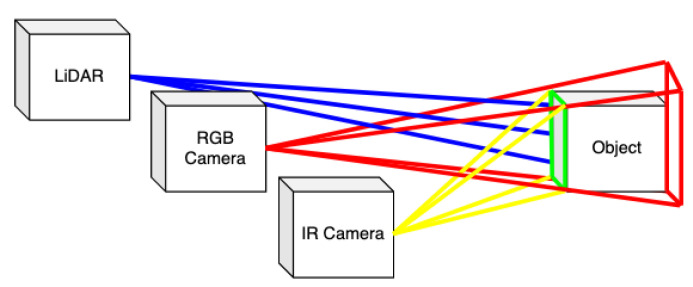
The basic principle of mapping detections from a RGB image into the thermal camera data. 1. (red) As the object is detected in the RGB image, it could be placed anywhere in the area bounded by the red frustum. 2. (blue) In the second step, we estimate the distance of the detected object from the camera by using the aggregated LiDAR measurements. 3. (green) Using the LiDAR measurements, we estimate the frontal face of the object. 4. (yellow) As the very last step, we can now reproject the estimated frontal face of the detected object to the IR camera.

**Figure 10 sensors-21-01552-f010:**
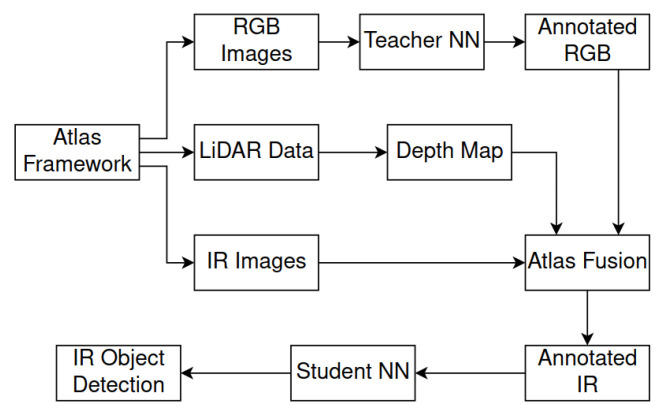
The scheme shows an overview of the entire method pipeline. Atlas Framework recorded the RGB camera, thermal camera, and LiDAR data. The teacher neural network annotated the RGB images, and the Atlas Fusion framework aggregated LiDAR data into the depth map. By combination of all this information, we were able to map detections from RGB into the thermal image. In this way, we created a dataset of 300,000 annotated images, and we used it to train the neural network to detect objects in the thermal image domain.

**Figure 11 sensors-21-01552-f011:**
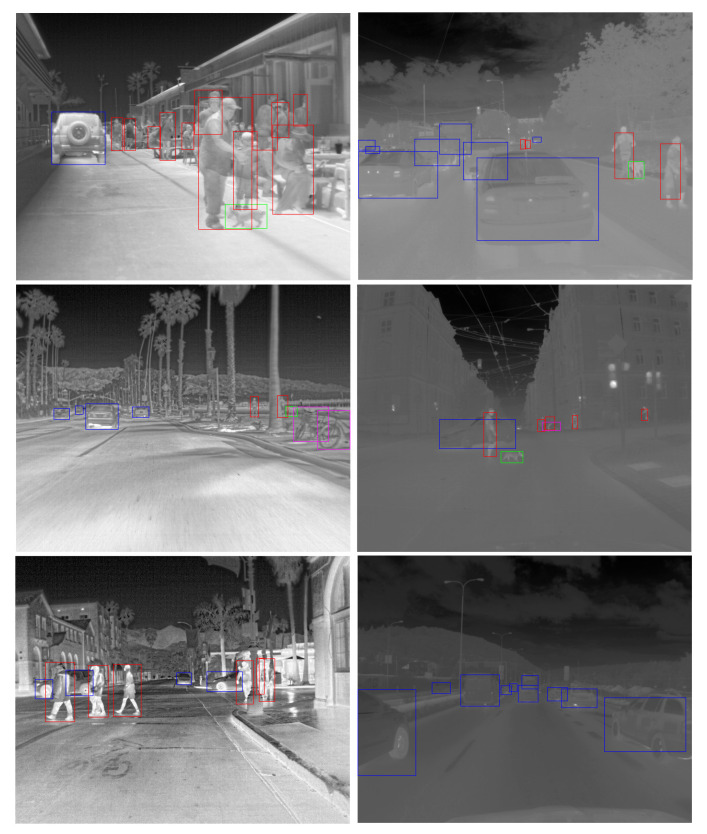
Examples from the FLIR and the BUDIR datasets. FLIR dataset is in the left column, BUDIR samples are in the right column. The bounding boxes visualize annotation: pedestrians—red, bikes—purple, vehicles—blue, animals—green.

**Figure 12 sensors-21-01552-f012:**
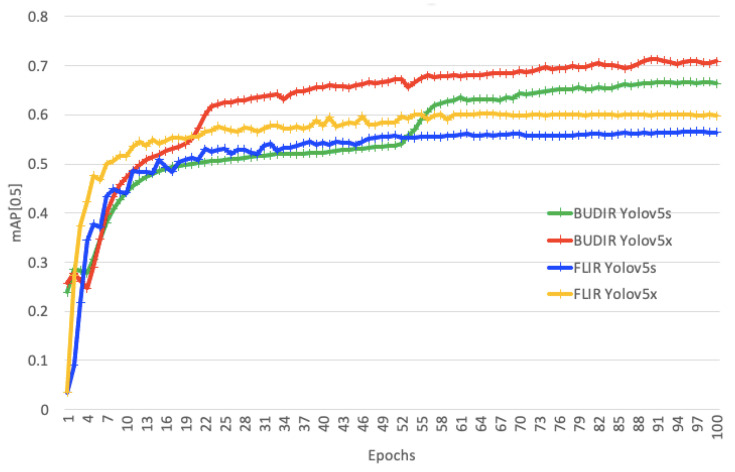
Learning curves of the YOLOv5s trained on FLIR data (blue), the YOLOv5x trained on FLIR data (yellow), YOLOv5s trained on BUDIR data (green), and the YOLOv5x trained on BUDIR data (red). As a metric, we used the mean Average Precision (mAP) (0.5) score.

**Figure 13 sensors-21-01552-f013:**
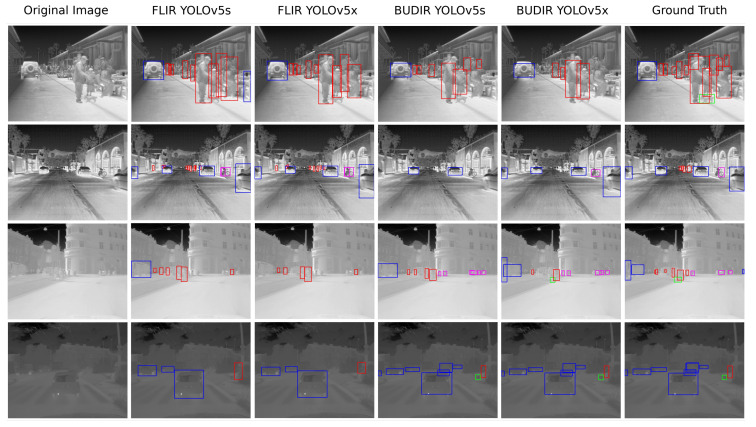
Image shows several examples of object detections performed by four neural networks. In columns from left to right: (1) original image, (2) YOLOv5s and FLIR, (3) YOLOv5x and FLIR, (4) YOLOv5s and BUDIR, (5) YOLOv5x and BUDIR, (6) ground truth. Annotation: pedestrians—red, bikes—purple, vehicles—blue, animals—green.

**Figure 14 sensors-21-01552-f014:**
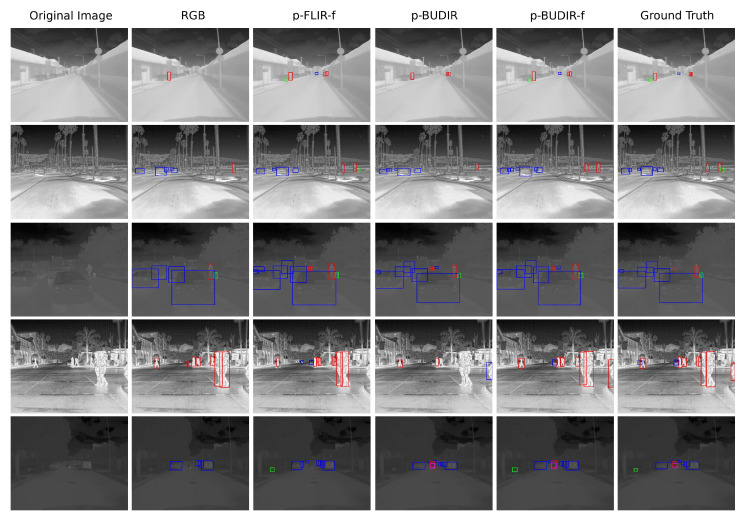
Figure shows the visual comparison of several neural networks performance. In columns from left to right: (1) original image, (2) RGB trained YOLOv5x, (3) RGB-pretrained, FLIR trained and hand-annotated finetunned YOLOv5x, (4) BUDIR-trained YOLOv5x, (5) RGB-pretrained and BUDIR-trained YOLOv5x, (6) RGB-pretrained, BUDIR-trained and hand-annotations finetuned YOLOv5x, (7) ground truth. Where p-prefix means RGB pretrained weights, and the f-suffix means fine-tuned on hand annotated data. Annotation: pedestrians—red, bikes—purple, vehicles—blue, animals—green.

**Table 1 sensors-21-01552-t001:** The table shows the number of class-specific annotations in the selected parts of the FLIR dataset.

Free FLIR ADAS Dataset				
	**Train**	**Validation**	**Video**	**Total**
Pedestrians	22,372	5779	21,965	50,116
Bikes	3986	471	1205	5662
Vehicles	41,260	5432	14,013	60,705
Animals	226	14	0	240

**Table 2 sensors-21-01552-t002:** An overview of the sensors installed on the ATLAS measurement platform.

Sensor	Type	Details	Freq.	Output Data
4× RGB camera	DFK33- -GX174	$75^{\circ }$ FoV 1920 × 1200 px	10	h265 video
Thermal camera	FLIR Tau 2	$70^{\circ }$ FoV 640 × 512 px	30	h265 video
2× LiDAR	Velodyne HDL-32e	32 beams ∼2000 pts/turn	10	point cloud
GNSS receiver	Trimble BX982	RTK accuracy Direct heading	20 20	global pose time
IMU	Xsens MTi-G-710	Combined non-electrical quantities sensor	400 400 100 400 400 400 50	accelerometer gyroscope magnetometer temperature global pose time pressure

**Table 3 sensors-21-01552-t003:** Number of annotations of the classes that are present in the synthetically generated data form Brno Urban Dataset (BUD)—IR.

Brno Urban Dataset-Automatically Annotated Dataset			
	**Training**	**Validation**	**Total**
Pedestrians	247,133	45,352	292,485
Vehicles	1,163,403	213,755	1,377,158
Bikes	15,904	3009	18,913
Animals	347	75	422

**Table 4 sensors-21-01552-t004:** Comparison of the YOLOv5s and YOLOv5x neural networks trained from scratch on the BUDIR and FLIR datasets and tested on the combined validation dataset.

YOLOv5s										
	**mAP (0.5)**		**mAP (0.5:0.95)**		**F1**		**Precisoin**		**Recall**	
**Training Dataset**	**FLIR**	**BUDIR**	**FLIR**	**BUDIR**	**FLIR**	**BUDIR**	**FLIR**	**BUDIR**	**FLIR**	**BUDIR**
Pedestrians	0.613	0.334	0.234	0.106	0.458	0.378	0.336	0.356	0.719	0.402
Bikes	0.525	0.317	0.185	0.133	0.343	0.308	0.231	0.258	0.665	0.382
Vehicles	0.643	0.718	0.312	0.370	0.556	0.622	0.477	0.544	0.666	0.727
Animals	0.126	0.556	0.126	0.256	0.196	0.559	0.236	0.554	0.168	0.564
**Average**	0.476	**0.481**	0.214	**0.216**	0.388	**0.467**	0.320	**0.428**	**0.555**	0.518
**YOLOv5x**										
	**mAP (0.5)**		**mAP (0.5:0.95)**		**F1**		**Precisoin**		**Recall**	
**Training Dataset**	**FLIR**	**BUDIR**	**FLIR**	**BUDIR**	**FLIR**	**BUDIR**	**FLIR**	**BUDIR**	**FLIR**	**BUDIR**
Pedestrians	0.621	0.451	0.253	0.141	0.488	0.476	0.366	0.459	0.733	0.495
Bikes	0.565	0.344	0.203	0.150	0.413	0.365	0.301	0.340	0.657	0.395
Vehicles	0.660	0.753	0.327	0.396	0.595	0.656	0.529	0.581	0.679	0.753
Animals	0.245	0.628	0.068	0.329	0.328	0.690	0.425	0.758	0.267	0.634
**Average**	0.522	**0.544**	0.213	**0.254**	0.456	**0.547**	0.405	**0.535**	**0.584**	0.569

**Table 5 sensors-21-01552-t005:** Comparison of the YOLOv5s and YOLOv5x neural networks trained, by using transfer learning and pre-trained original RGB-trained weights, on the BUDIR and FLIR datasets and tested on the combined validation dataset. In addition, we have added a comparison with original RGB-trained weights.

YOLOv5s															
	**mAP (0.5)**			**mAP (0.5:0.95)**			**F1**			**Precisoin**			**Recall**		
**Train. Dataset**	**FLIR**	**BUDIR**	**RGB**	**FLIR**	**BUDIR**	**RGB**	**FLIR**	**BUDIR**	**RGB**	**FLIR**	**BUDIR**	**RGB**	**FLIR**	**BUDIR**	**RGB**
Pedestrians	0.611	0.344	0.609	0.240	0.144	0.282	0.453	0.392	0.484	0.328	0.410	0.379	0.731	0.376	0.671
Bikes	0.582	0.315	0.326	0.196	0.139	0.135	0.426	0.338	0.353	0.311	0.304	0.333	0.676	0.380	0.375
Vehicles	0.658	0.717	0.550	0.319	0.381	0.251	0.581	0.630	0.459	0.509	0.561	0.372	0.677	0.718	0.598
Animals	0.183	0.644	0.088	0.048	0.275	0.029	0.265	0.675	0.161	0.264	0.733	0.247	0.266	0.625	0.119
Average	0.509	0.505	0.393	0.201	0.227	0.174	0.431	0.509	0.364	0.353	0.502	0.333	0.588	0.525	0.441
**YOLOv5x**															
	**mAP (0.5)**			**mAP (0.5:0.95)**			**F1**			**Precisoin**			**Recall**		
**Train. Dataset**	**FLIR**	**BUDIR**	**RGB**	**FLIR**	**BUDIR**	**RGB**	**FLIR**	**BUDIR**	**RGB**	**FLIR**	**BUDIR**	**RGB**	**FLIR**	**BUDIR**	**RGB**
Pedestrians	0.652	0.380	0.686	0.268	0.117	0.349	0.492	0.432	0.541	0.360	0.446	0.422	0.777	0.419	0.755
Bikes	0.617	0.380	0.389	0.234	0.138	0.153	0.423	0.355	0.391	0.298	0.301	0.320	0.726	0.434	0.503
Vehicles	0.672	0.742	0.629	0.325	0.390	0.294	0.572	0.647	0.518	0.485	0.570	0.423	0.699	0.749	0.667
Animals	0.267	0.678	0.344	0.069	0.357	0.119	0.279	0.755	0.443	0.244	0.861	0.539	0.327	0.673	0.376
Average	0.552	0.545	0.512	0.224	0.251	0.229	0.442	0.548	0.473	0.347	0.545	0.426	0.632	0.569	0.575

**Table 6 sensors-21-01552-t006:** The table shows the comparison of all neural networks that we trained for this paper’s purpose, measured by mAP (0.5) metrics. The p-prefix means neural networks with pre-trained weights on RGB images. The f-suffix expresses the finetuned weights by the 150 hand-annotated images.

mAP (0.5)												
	**YOLOv5x**							**YOLOv5s**				
**Train. Dataset**	**RGB**	**FLIR**	**p-FLIR**	**p-FLIR-f**	**BUDIR**	**p-BUDIR**	**p-BUDIR-f**	**RGB**	**FLIR**	**p-FLIR**	**BUDIR**	**p-BUDIR**
Pedestrians	0.686	0.621	0.652	0.629	0.451	0.380	0.729	0.609	0.613	0.611	0.334	0.344
Bikes	0.389	0.565	0.617	0.599	0.344	0.380	0.592	0.326	0.525	0.582	0.317	0.315
Vehicles	0.629	0.660	0.672	0.701	0.753	0.742	0.780	0.550	0.643	0.658	0.718	0.717
Animals	0.344	0.245	0.267	0.290	0.628	0.678	0.543	0.088	0.126	0.183	0.556	0.644
Average	0.512	0.523	0.552	0.555	0.544	0.545	0.661	0.393	0.477	0.509	0.481	0.505

## Data Availability

Brno Urban Dataset: https://github.com/Robotics-BUT/Brno-Urban-Dataset (accessed on 23 February 2021); Software used for BUDIR dataset creation: https://github.com/Robotics-BUT/Atlas-Fusion (accessed on 23 February 2021).

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
