# Peer review of "Fully Automated DCNN-Based Thermal Images Annotation Using Neural Network Pretrained on RGB Data"

_sensors, 2021, doi:10.3390/s21041552_

Round 1

Reviewer 1 Report

The paper deals with automatic annotation of thermal images in the filed of autonomous driving according to the used datasets. The paper follows standard way of scientific reporting (with introduction, related work, proposed approach, evaluation and conclusion). However, I found number of issues which must be resolved before judging about paper acceptance:

Introduction does not contain precise statement of paper contribution and how it actually adds knowledge or supplement current state-of-the-art. I suggest authors to rewritten the Introduction (summarize similar approaches to transfer learning or domain shift and point out paper contribution).

The related work is not written at satisfactory level. The authors should summarize similar ideas based on deep learning which are dealing with domain shift (domain adaptation).

The language is sometimes insufficiently accurate or too colloquial for scientific text. Just several examples:
Line 51
Figure 2 caption: These two sensors are the most important in this paper.
Figure 1 caption: right top - 3D point cloud model of the surrounding
Abstract: "This paper presents our method of transferring the object detection ability from the neural network trained on the RGB image dataset to the neural network we use to detect thermal images."

There is lot of surplus information in text. For example, precision, recall and F1 measure definitions, definition of TP, FP and FN... This are well known metrics in the field of computer vision i.e. object detection. This part can be shorten.

In my opinion the results section does not support to the required degree a declared contribution of this paper. There is no comparison with other methods in the field of domain adaptation.

Author Response

Thank you very much for your review. For my response, please see the appended pdf document.

Best regards,
Adam Ligocki

Reviewer 2 Report

  1. Lines 129-130: Parameters need to be explained.
  2. It can be seen from the text that the number of certain categories is very small, such as animals. Do you consider the imbalance between balanced categories? If so, please describe briefly.
  3. Lines 258-264: According to the description, there is a question on the composition of the test dataset. You are taking out some pictures from the validation set and combining them into a test dataset, so whether these pictures participate in the training of the model. If this part of the picture is involved in the training of the model, strictly speaking, the test dataset is not rigorous; if not, please correct the description.
  4. When I first saw the title of the article, I thought of model compression or knowledge distillation. After reading the article carefully, the core content is transfer learning. I think whether you should revise the title of the article and directly highlight the core elements.
  5. Lines 143-150: Atlas fusion sounds very interesting, and it is also the core of unsupervised data augmentation. More details should be considered.
  6. The writing style of this paper needs to be improved and the language needs to be concise.

Author Response

(The authors gave the same response as above.)

Round 2

Reviewer 1 Report

I started to read the paper and I found several issues in the Introduction and Datasets which discouraged me to read the whole paper before authors improve it. Writing style should be improved to successfully report what is actually done in the paper.

I attached the comments for the part I read (marked with the blue color).

Author Response

Dear Referee,
Thank you for your notes in the second review. We focused mainly on making the text more clear and better to understand. We removed the redundant sentences and rephrased number of unclear formulations. However, the overall structure of the text stays the same compared to the previous version.
We appreciate your efforts related to the manuscript and hope that the alterations have increased its overall quality and acceptability. Thank you for considering the revised version of the article as regards possible publication.

Best regards,
Adam Ligocki
Corresponding author 
